# Cross-Linked Enzyme Aggregate (CLEA) Preparation from Waste Activated Sludge

**DOI:** 10.3390/microorganisms11081902

**Published:** 2023-07-27

**Authors:** Ziyi Liu, Stephen R. Smith

**Affiliations:** Department of Civil and Environmental Engineering, Imperial College London, South Kensington Campus, London SW7 2AZ, UK

**Keywords:** activated sludge, enzyme, cross-linked enzyme aggregate (CLEA), resource recovery, wastewater treatment

## Abstract

Enzymes are used extensively as industrial bio-catalysts in various manufacturing and processing sectors. However, commercial enzymes are expensive in part due to the high cost of the nutrient medium for the biomass culture. Activated sludge (AS) is a waste product of biological wastewater treatment and consists of microbial biomass that degrades organic matter by producing substantial quantities of hydrolytic enzymes. Recovering enzymes from AS therefore offers a potential alternative to conventional production techniques. A carrier-free, cross-linked enzyme aggregate (CLEA) was produced from crude AS enzyme extract for the first time. A major advantage of the CLEA is the combined immobilization, purification, and stabilization of the crude enzymes into a single step, thereby avoiding large amounts of inert carriers in the final enzyme product. The AS CLEA contained a variety of hydrolytic enzymes and demonstrated high potential for the bio-conversion of complex organic substrates.

## 1. Introduction

Activated sludge (AS) mainly consists of microbial biomass that degrades organic matter in wastewater by producing substantial quantities of hydrolytic enzymes. Thus, AS is a potentially cost-effective alternative crude biomass source for hydrolytic enzyme extraction when compared to conventional enzyme production methods [1]. Protease and glycosidase (e.g., amylases and cellulases) are particularly abundant in crude AS extracts according to our previous research [2] and are also the main groups of enzymes used in industrial hydrolysis reactions [3]. However, crude enzymes extracted from AS are diluted in the extraction medium, and their storage and operational stability are relatively limited, thus restricting their potential industrial application [4].

These problems may be avoided by upgrading crude AS extracts to immobilize, consolidate, and stabilize the hydrolytic enzymes, which also increase enzyme resistance to denaturation, enhance mechanical strength, and facilitate recycling of the enzyme catalyst within an industrial reaction process. Unfortunately, conventional, carrier-bound enzyme immobilization methods also introduce large amounts of non-catalytic material into the enzyme product [5]. However, this can be avoided, and these desirable properties may be gained in a single process by producing a cross-linked enzyme aggregate (CLEA) from AS enzyme extracts.

Enzyme cross-linking involves two key steps: (1) aggregation of the soluble enzyme by protein precipitation from an aqueous solution with an inorganic salt (e.g., ammonium sulfate ((NH_4_)_2_SO_4_)) or organic solvent (e.g., acetone), followed by, (2) chemical cross-linking of the aggregated enzyme by Schiff base (imine) formation from the nucleophilic reaction by amino groups of the enzyme with a bifunctional reagent, i.e., the cross-linker [6]: (Enzyme)-NH_3_^+^ + OHC-(Cross-linker) → (Enzyme)-N=CH-(Cross-linker)(1)

Glutaraldehyde contains two aldehyde residues and is one of the most widely used cross-linker reagents for CLEA preparation due to its strong reactivity towards protein moieties [7]. However, in some instances, glutaraldehyde cross-linking can alter the active center of the enzyme to cause severe inactivation [8]. Therefore, alternative, macro-molecular cross-linkers (e.g., dextran aldehyde) have been proposed, which react less aggressively with, and, thus, are less damaging to the enzyme molecular structure compared to glutaraldehyde [9]. Cross-linked enzyme aggregate preparation also depends on the abundance of reactive lysine residues at the surface of the enzyme to interact with the cross-linker. If there are insufficient reactive residues, the enzyme can be co-aggregated with an inert protein additive (e.g., bovine serum albumin (BSA)) to enhance CLEA formation [10].

The majority of previous studies prepare CLEAs from purified, commercially available enzyme sources (e.g., [11,12,13,14,15]), and there has only been limited investigation of CLEA production from crude, enzyme rich biomass (e.g., fermentation broths of microorganisms [16], mung beans [17], and fresh fruit [18]). Using the crude enzyme extract from AS for CLEA preparation has not been previously reported as far as we are aware, and this would represent a new approach to formulating AS enzymes into an industrial bio-catalyst product. In addition, the co-immobilization of different enzymes released from crude biomass sources (e.g., AS) in a single aggregate to produce “multi-CLEAs” [19] may be particularly valuable for multi-purpose applications to catalyze several unrelated reactions simultaneously in an industrial process. Thus, the performance of AS multi-CLEA for the bio-conversion of complex organic substrates should be assessed.

The aims of this study, therefore, included the following: (1) develop a viable CLEA product from crude AS enzyme extracts; (2) determine the enzyme activity recovery rate and chemical bond formation of the AS CLEA; and (3) hydrolyze complex organic substrates using the AS CLEA, including wheat flour, as a representative complex organic substrate source, and waste activated sludge (WAS) from the biological wastewater treatment process, as a potential pretreatment technique, for example, for anaerobic digestion [20,21,22].

## 2. Materials and Methods

### 2.1. Materials

Waste-activated sludge samples (total solids (TS) = 5–7%, volatile solids (VS) = 4–5% TS (as received), pH = 6.9) were collected from the thickening belt after flocculant dosing (Flopam, 0.24% *w*/*w* active) at a major municipal sewage treatment plant in the UK.

Sludge samples were transported to the lab in an insulated cool box containing freezer packs on the day of collection and stored in a fridge overnight at 4 °C. Crude enzyme extractions and other assays were performed the following day, unless otherwise specified.

Food-grade white wheat flour was obtained from a major UK supermarket retailer with the following composition: TS, 93.0%; VS, 92.0% and the TS for the content of lipids, carbohydrates, fiber, and proteins was equivalent to 1.7, 71.7, 3.3, and 12.7% TS, respectively.

### 2.2. CLEA Preparation

#### 2.2.1. General Processes

Crude enzyme extract from AS samples was prepared following the procedure described by Liu and Smith [2]: AS (10 g VS/L fresh weight) was disrupted using sonication (energy intensity of 872 W/L and 10 min duration) and surfactant treatment (TX100 dose equivalent to 1% *v*/*v*, stirred at 120 rpm for 45 min), followed by solid–liquid separation by centrifugation (12,000× *g* for 15 min); the supernatant was collected as the crude enzyme extract.

The crude enzyme extract was mixed with a lysine-rich additive (BSA), followed by a precipitant ((NH_4_)_2_SO_4_ or acetone (procedure described in Section 2.2.2)), and it was continuously stirred by magnetic stirrer for 30 min in an ice-water bath. A cross-linker (glutaraldehyde or dextran aldehyde, preparation methods described in Section 2.2.3) was subsequently added in a drop-wise manner to the required concentration. The mixture was continuously stirred for 3.5 h at room temperature and centrifuged at 10,000× *g* (10 min, 4 °C). The sediment was washed three times and re-suspended in Tris-HCl buffer (10 mM, pH 7.0) to the original volume. Prepared CLEA suspensions were stored at 4 °C in a fridge before use. The volume activity of the CLEAs was measured to compare the enzyme activity recovery rate after cross-linking. The *R_CLEA_* (%), is expressed as follows: (2)$$R_{CLEA}\%=\frac{Activity\;in\;the\;CLEA}{Activity\;of\;the\;original\;crude\;enzyme\;solution}\times 100$$

#### 2.2.2. Chemical Precipitation of Soluble Enzymes

An inorganic ((NH_4_)_2_SO_4_) and an organic (acetone) protein precipitant were evaluated for enzyme precipitation through the following procedures.

Solid (NH_4_)_2_SO_4_ (oven-dried at 107 °C overnight and cooled in a desiccator) was slowly added into the crude enzyme solution to achieve 80% saturation; the weight of solid (NH_4_)_2_SO_4_ required per L of solution was 516 g [23]. (NH_4_)_2_SO_4_ was added with continuous stirring in an ice-water bath to avoid localized high concentrations, which could potentially cause enzyme denaturation [24].

Cold acetone (Sigma-Aldrich, analytical grade, stored at −20 °C) was added to the enzyme solution at a volume ratio (acetone: enzyme solution) equivalent to 4:1 by continuous stirring for 30 min in an ice-water bath; the volume ratio (4:1) was adopted to obtain complete precipitation and avoid enzyme inactivation [25,26]. Mixing organic solvents into aqueous solutions is an exothermic process; therefore, the addition of cold acetone reduced the potential heat generation and the risk of protein denaturation.

#### 2.2.3. Cross-Linking of Precipitated Enzymes

The effectiveness of two chemical cross-linkers was examined, including: (1) glutaraldehyde, prepared by diluting a proprietary chemical (Sigma-Aldrich, 50% *w*/*v* aqueous solution) with reverse osmosis (RO) water to 5% *w*/*v*; and (2) dextran aldehyde, a macro-molecular poly-aldehyde prepared from dextran via periodate oxidation, as described by Mateo et al. [27]. Dextran (average MW = 150,000 Da), 1.65 g, was dissolved in 50 mL of RO water, and 3.85 g sodium meta-periodate was added. The solution was stirred at room temperature for 90 min and purified by five dialysis cycles, each for 2 h, against 5 L RO water at room temperature using dialysis tubing (Sigma-Aldrich, MW cut-off = 12,400 Da, average diameter = 20 mm, capacity equivalent to 328 mL/m). The final volume of the dextran aldehyde was approximately 56 mL. The periodate oxidation process was performed in the dark to prevent the photolysis of periodate [28].

### 2.3. CLEA Preparation from Crude AS Enzyme Extracts

#### 2.3.1. Enzyme Cross-Linking with Glutaraldehyde

Crude AS enzyme extract was mixed with BSA (5 mg/mL) and subjected to the CLEA preparation process by precipitation with 80% saturated (NH_4_)_2_SO_4_ at 0 °C for 30 min and cross-linking with increasing concentrations of glutaraldehyde (from 0.005 to 2.0%) at room temperature for 3.5 h. The effect of glutaraldehyde concentration on CLEA preparation was quantified using amylase as an indicator enzyme, since this enzyme presented the largest overall activity detected in crude AS extracts (see [2]). The activities of amylase and of the other main target enzyme types, which included—protease, lipase, and cellulase—were subsequently measured at the glutaraldehyde concentration that yielded the maximum amylase activity.

#### 2.3.2. Enzyme Cross-Linking with Dextran Aldehyde

Protein precipitants ((NH_4_)_2_SO_4_ or acetone) were added separately to crude BSA (5 mg/mL)-amended AS extracts following the procedures described in Section 2.2.2. Cross-linking of the precipitated enzymes was performed with dextran aldehyde at room temperature for 3.5 h. The concentrations of dextran aldehyde were in the range of 0.01 to 0.28% *w*/*v* for (NH_4_)_2_SO_4_ and in the range of 0.01 to 0.20% *w*/*v* in the case of acetone. The activities of amylase protease, lipase, and cellulase were measured before and after reaction with the cross-linker.

### 2.4. Enzyme Activity Assays

Enzyme activity was measured based on the rate of hydrolysis reaction product formation [29]. Thus, one enzyme activity unit (U) was equivalent to the formation of 1 μmol of product per minute under the assay conditions.

Amylase activity was determined by the 3,5-dinitrosalicylic acid (DNS) reaction [4], with starch (2% *w*/*w*, in 20 mM sodium phosphate buffer with 6.7 mM NaCl, pH 6.9) as the substrate and glucose as the standard. Protease activity was measured following the Lowry method described by Nabarlatz et al. [30] using casein (0.65% *w*/*v*, prepared in 50 mM phosphate buffer, pH 7.5) and L-tyrosine as the substrate and standard, respectively, which was modified according to Potty [31] to avoid interference from phenolic compounds (that could be potentially co-extracted from the sludge matrix) and Tris buffer (used to extract crude enzymes, see Section 2.2.1). Lipase activity was determined using p-nitrophenol palmitate (pNPP) as substrate (16.5 mM pNPP in iso-propanol mixed with 50 mM Tris-HCl buffer containing 0.4% *w*/*v* TX100 and 0.1% *w*/*v* Arabic gum, pH 8.0, at a volume ratio = 1:9), and the release of p-nitrophenol (pNP) was measured continuously by spectrophotometric determination [32]. Cellulase activity was measured according to the carboxymethyl cellulose (CMC) method [33] using CMC as the substrate (2% *w*/*w*, in 50 mM citrate buffer, pH 4.8) and a glucose standard.

### 2.5. Fourier-Transform Infrared Spectroscopy

Fourier-transform infrared (FTIR) spectrophotometry (Magna 560, Nicolet Instrument Corporation, Madison, WI, USA) was used to examine the following: (1) the oxidation of dextran into dextran aldehyde (data shown in Appendix A, Figure A1) and (2) the formation of new chemical bonds after cross-linking AS enzymes under optimum preparation conditions.

### 2.6. CLEA Morphology

The morphology of the soluble enzyme and immobilized CLEA was examined by scanning electron microscopy (SEM; Sigma 500 VP Field Emission-Scanning Electron Microscope, Zeiss, Oberkochen, Germany). Crude enzyme extract and CLEA produced under optimum conditions were prepared for SEM by freeze-drying for three consecutive days in a freeze-dryer (Modulyo 4K; Edwards, Burgess Hill, UK).

### 2.7. Hydrolysis of Organic Materials by CLEA

#### 2.7.1. Substrate Preparation

The hydrolytic activity of AS CLEA was examined using wheat flour and WAS as complex organic substrates.

Wheat flour is a refined organic matrix containing carbohydrates (mainly in the form of starch), proteins, and lipids and is a widely used substrate to quantify the activity of hydrolytic enzymatic reaction systems (e.g., [34]). The wheat flour was prepared by mixing with RO water to a TS content of 1.5% *w*/*w*. The suspension was freshly prepared and continuously stirred at 100 rpm by a magnetic stirrer prior to CLEA hydrolysis.

Waste-activated sludge also has a complex macro-molecular composition of proteins, lipids, and carbohydrates, but it is structurally and micro-biologically more dynamic than flour, as it also contains active microbial cells, their metabolites, and extra-cellular polymeric substances (EPS). Samples of thickened WAS were diluted 5 times with RO water and received the following pre-treatments before enzymatic hydrolysis to disrupt the endogenous bio-activity of sludge cells and reduce potential consumption and interference with detecting hydrolysis products [35,36]: Gamma (γ) irradiation: diluted WAS was exposed to a cesium (Cs) 137 radiator (Gammacell 3000 ELAN, Best Theratronics Ltd., Canada). The WAS suspension (450 mL) was placed in a 500 mL HDPE bottle and was exposed to the radiator for 10 h at room temperature; the total radiation dose was 1944 Gy at a dose rate of 3.24 Gy/min.Sonication: diluted WAS was treated by sonication probe (VCX130, Sonics & Materials Inc., Newtown, CT, USA) for 10 min duration at 0 °C, 40% amplitude, in pulse cycle mode with 1 min on followed by 1 min off.

Diluted whole WAS (without biomass disruption), as well as sonicated and γ-irradiated samples, were stored in a fridge at 4 °C, and CLEA hydrolysis experiments on these substrates were conducted the following day. The biological activity of disrupted sludge samples was examined (data not shown) by a standard culturing method [37] and was significantly reduced, albeit not completely eliminated, compared to undisrupted sludge.

#### 2.7.2. Hydrolysis of Organic Substrates

Prepared substrate (200 mL) was transferred to a 500 mL Schott bottle and pre-heated in a water bath at 40 °C for 10 min. CLEA suspension was prepared following the optimum procedure by acetone precipitation and dextran aldehyde cross-linking based on the enzyme activity results, which are described in Section 3.2.2. The activity of the CLEA suspension was 3.024 ± 0.004 and 0.103 ± 4 × 10^−5^ U/mL for amylase and protease, respectively. Flour and WAS were added to the CLEA suspension on a CLEA/substrate volume ratio equivalent to 1:100 and 1:10 for flour hydrolysis, as well as 1:10 for WAS hydrolysis. The CLEA/substrate mixture was stirred continuously at 120 rpm and maintained at 40 °C for 26 h. Similar flasks were prepared for the control with an equivalent volume of Tris-HCl buffer (10 mM pH 7.0) in place of the CLEA suspension. Aliquots (20 mL) were collected from each bottle immediately and after 1, 2, 3.5, 4.5, 5.5, 6.5, 24, and 26 h of incubation and heated in boiling water for 10 min to inactivate the CLEA. The boiled mixture was cooled immediately using running tap water and centrifuged at 18,000 rpm for 20 min. The supernatant was collected and filtered through a cellulose nitrate membrane filter paper (Whatman NC45, 0.45 μm pore size, 25 mm diameter) to remove residual cell debris and particulate matter, and it was stored in a freezer at −18 °C prior to further testing.

#### 2.7.3. Indicators of the Progress of Enzymatic Hydrolysis


Soluble Total Organic Carbon


The soluble total organic carbon (sTOC) concentration in the filtrates of reaction mixtures (see Section 2.7.2) was measured with an automatic TOC analyzer (Shimadzu TOC-VWP analyzer, Shimadzu Corporation, Kyoto, Japan) to provide a general indication of enzymatic solubilization of particulate substrates in flour and WAS. The sTOC content was quantified using a calibration relationship with potassium hydrogen phthalate as the standard.
Reducing Sugar and Tyrosine

Starch and proteins constitute the majority of the substrates by mass in wheat flour (>80% *w*/*w*); therefore, the hydrolysis products of amylase and protease enzymatic activity in the CLEA, of reducing sugar and tyrosine, respectively, were measured to show the effects on flour hydrolysis.

More than 60% of the TS content of WAS is composed of various polysaccharides and proteins [38]. For hydrolysis of whole WAS, reducing sugar accumulation was, therefore, measured as an indicator of the performance of polysaccharide-degrading enzymes in the CLEA (including amylase and cellulase). However, the change in reducing sugar content for whole WAS showed only a slight, non-significant (*p* < 0.05) increase (data not shown). Therefore, in subsequent experiments with γ-irradiated and sonicated WAS, both tyrosine and reducing sugar accumulation were measured. The background concentrations of the sTOC, reducing sugar, and tyrosine in the Tris-HCl buffer (10 mM, pH 7.0) used to prepare the CLEA suspension (Section 2.2.1) were measured and deducted from the results observed after hydrolysis to normalize the data and minimize potential internal interferences.

#### 2.7.4. Progress Curve of Enzymatic Hydrolysis

Hydrolysis reaction product concentrations can be described by the following first-order function [39,40]: (3)$$C=C_{0}+(C_{\infty }{-C}_{0})(1-e^{-kt}),$$
where *C*_0_ and *C* are the concentrations (mg/g TS) of hydrolysis product initially and after reaction time, *t* (h), respectively, *C*_∞_ is the equilibrium concentration (mg/g TS) of the product, and *k* denotes the solubilization rate constant (h^−1^). The parameters *C*_0_, *C*_∞_, and *k*, for accumulated reducing sugar, tyrosine, and sTOC were estimated by fitting the model to the experimental data using the Graphpad Prism 7.0 software within 95% confidence intervals. The statistical validation protocol is described in Appendix B.

The reactions did not proceed to final product equilibrium in the enzymatic hydrolysis experiments for various reasons discussed later. Therefore, the first-order relationships were used for indicative purposes to estimate the maximum potential yields (% TS) of the sTOC, reducing sugar, and tyrosine from the enzymatic hydrolysis of wheat flour based on the following equation and mass balance procedure: (4)$$Yield(\%\;TS)=\frac{C_{\infty }-C_{0}}{C_{Total}-C_{0}}\times 100$$
*C_Total_* for sTOC was derived from the specific VS content (0.989 g/g TS) and the element composition ratio of wheat flour (C:H:O:N:S = 1:2.004:0.625:0.249:0.01) reported by [41], and it was equivalent to 0.568 g/g TS.*C_Total_* for reducing sugar was derived by dividing the total carbohydrate content in the flour (0.771 g/g TS, which was assumed to be in the form of starch) by the coefficient, 0.9 [42], and it was equivalent to 0.857 g/g TS.*C_Total_* for tyrosine was estimated based on the value reported by Siddiqi et al. [43], and it was 5.6 mg/g TS.

Equation (4) was also used to estimate the maximum potential yield of tyrosine released by CLEA hydrolysis of sonicated and γ-irradiated WAS. Vriens et al. [44] reported that tyrosine represented 2.4% of crude protein (N × 6.25) in WAS, and the N content was 9.6–11.2% *w*/*w* TS. Thus, *C_Total_* for tyrosine in WAS was assumed to be in the range 14.4–16.8 mg/g TS, and a representative value of 15.6 mg/g TS was adopted in the total yield calculation for this product from WAS hydrolysis.

### 2.8. Statistical Analysis

Where applicable, data are shown as the mean value of three replicates ± standard deviation, unless otherwise stated. One-way analysis of variance was used to determine statistical significance at a *p* value of ≤0.05. All statistical analyses were performed in Microsoft Excel software, unless otherwise stated.

## 3. Results and Discussion

### 3.1. Cross-Linking AS Enzymes with Glutaraldehyde

The effect of glutaraldehyde concentration on the amylase activity of the CLEAs prepared with BSA addition (5 mg/mL) and precipitation with 80% saturated (NH_4_)_2_SO_4_ is shown in Figure 1. The amylase activity in the CLEA increased significantly (*p* < 0.05) with decreasing concentrations of glutaraldehyde to 0.04%, but it remained relatively consistent below this level (0.04–0.005% *w*/*v*). A slight improvement in activity, to a value equivalent to 19.0% ± 0.037 of the crude enzyme extract, was obtained at the smallest concentration of glutaraldehyde, but the overall mean activity for the concentration range of 0.04–0.005% *w*/*v* was equivalent to 17.6% ± 0.86. No amylase activity was detected in the concentration range of 0.3–2.0% *w*/*v*, thus indicating the complete inactivation of the enzyme at higher doses of glutaraldehyde. The partial/full loss of enzyme activity in the AS CLEA compared to the crude extract can be explained by changes in enzyme morphology (Appendix C) and the reduced availability of the catalytically active sites after cross-linking. Regions within the interior of the CLEA structure may also have been less accessible to substrate molecules, and this is likely to be more pronounced for amorphous forms (Appendix C, Figure A2c) [45].

The results indicated that optimum cross-linking occurred at a glutaraldehyde concentration of 0.04% *w*/*v*, which yielded an amylase activity in the maximum range. The activity recovery rate (relative to the crude enzyme extract, *R_CLEA_*) of the lipase, protease and cellulase in the CLEA was also determined. The rate was 23.6% for lipase; however, protease and cellulase activity were not detected.

### 3.2. Cross-Linking AS Enzymes with Dextran Aldehyde

#### 3.2.1. Ammonium Sulfate Precipitation

The effect of dextran aldehyde concentration on the enzymatic activity of CLEAs prepared in conjunction with enzyme precipitation with 80% saturated (NH_4_)_2_SO_4_ and 5 mg/mL of BSA addition is shown in Figure 2a. The highest amylase activity recovery, equivalent to 27.5% relative to the crude enzyme solution, was obtained at a dextran aldehyde concentration equivalent to 0.08% *w*/*v*, thus representing an increase of 57% in the enzyme activity compared to the mean upper activity observed with glutaraldehyde. This was consistent with the frequently reported increase in enzyme recovery rates obtained with dextran compared to glutaraldehyde [46]. In contrast to glutaraldehyde, protease activity was also detected in the CLEA prepared by dextran aldehyde cross-linking, with a recovery rate in the range of 0–25.7% (at dextran aldehyde concentration of 0.04–0.28% *w*/*v*) compared to the crude AS enzyme extract. The improved activity obtained by dextran aldehyde cross-linking may be explained by the larger molecular size compared to glutaraldehyde (MW = 150,000 Da vs 100.11 Da for glutaraldehyde), which limits the potential penetration of the cross-linker into enzyme molecules and the reaction with active sites or essential amino acid residues of the enzyme [27]. In addition, dextran molecules have multiple aldehyde groups that offer more reaction sites when compared to the two available in glutaraldehyde monomers. It is also more suitable for cross-linking enzymes with essential functional groups at the catalytic active center (e.g., lysine) that are susceptible to binding with small cross-linker molecules. Consequently, dextran aldehyde is less aggressive for amylase cross-linking compared to glutaraldehyde. Another advantage of dextran aldehyde is that it is a non-toxic renewable poly-saccharide macro-molecular polymer [47].

Thus, dextran aldehyde concentrations above the optimum value had less severe impacts on enzyme activity compared to glutaraldehyde under equivalent conditions (Figure 1 and Figure 2a); nevertheless, increasing the dextran aldehyde up to a maximum concentration equivalent to 0.28% *w*/*v* reduced the amylase recovery rate to approximately 20.1% (Figure 2a). The effects of increasing cross-linker concentration on the activities of the protease and lipase were more dynamic than for amylase. Thus, the protease activity was undetectable at dextran aldehyde concentrations of 0.01% and 0.04% *w*/*v*, but increased to 20.4% and 25.7% at concentrations of 0.08% and 0.12% *w*/*v*, respectively. However, the activity was reduced by approximately 50% at the largest dextran aldehyde concentrations used (0.24 and 0.28% *w*/*v*) compared to the maximum observed activity rate. A similar behavior was found for lipase, which also showed the highest activity recovery rate (26.3%) at a concentration of 0.12% *w*/*v* of dextran aldehyde. Therefore, although it was generally a less aggressive cross-linking agent compared to glutaraldehyde, high concentrations of dextran aldehyde may also damage enzymes and reduce the overall activity recovery. Thus, it is critical to optimize the reaction conditions to maximize the efficiency of the enzyme activity recovery from AS, as well as from other biomass types.

#### 3.2.2. Acetone Precipitation

Figure 2b shows the effect of dextran aldehyde concentration on the enzymatic activity of the CLEAs prepared with 4:1 acetone precipitation and 5 mg/mL of BSA addition. The activities of the target enzymes markedly increased with acetone precipitation by up to approximately two times the activity rates obtained with NH_4_(SO_4_)_2_ (Figure 2a). The maximum amylase, protease, and lipase activities in the CLEA with acetone precipitation were obtained at a dextran aldehyde concentration of 0.04% *w*/*v* and were equivalent to 42.6%, 48.3%, and 54.4%, respectively. Upon increasing the dextran aldehyde concentration further, up to a maximum value of 0.2% *w*/*v*, this reduced the activities of the different enzymes to varying degrees. Under this condition, amylase and protease were completely inactivated, and the lipase activity was reduced by approximately 70%. By contrast, the activity recovery of cellulase was generally not significantly (*p* > 0.05) affected by the dextran aldehyde concentration above 0.04% *w*/*v* (note the cellulase sample at 0.16% *w*/*v* was compromised, and this result is not reported) and was in the range of 31.0 to 35.1%. This behavior was in contrast to (NH_4_)_2_SO_4_ precipitation, where no cellulase activity was detected.

The decline in enzyme activity with (NH_4_)_2_SO_4_ compared to acetone precipitation could be explained due to the introduction of a high level of charged, ionic chemical species into the enzyme solution, thus causing the excessive replacement of water and the loss of the hydration layer from enzyme molecules [48]. Essential hydrophobic, or at least non-hydrophilic, poly-peptide chains of enzymes may, therefore, become exposed to the surrounding water, which may lead to detrimental conformational changes of the enzyme protein structure [49]. In addition, inhibitory materials potentially present in crude enzyme extracts, such as those obtained from complex matrices such as AS, which may also hinder enzyme aggregation by (NH_4_)_2_SO_4_ [16].

The largest overall enzyme activity recovery rates in the CLEAs prepared from crude AS enzyme extracts were, therefore, obtained with acetone precipitation (volume ratio = 4:1, 0.5 h at 0 °C), dextran aldehyde cross-linking (0.04% *w*/*v*, 3.5 h at room temperature), and 5 mg/mL of BSA addition. These conditions are, therefore, recommended to produce CLEAs from the crude enzyme extracts of AS.

### 3.3. Fourier-Transform Infrared Spectroscopy Analysis

The Fourier-transform infrared spectra (Figure 3) showed notable conformational changes in the secondary structure of the AS enzymes after CLEA preparation, thereby confirming, and being consistent with, the occurrence of cross-linking reactions. The presence of typical peaks for protein polypeptides (amides) were observed for both the crude, soluble AS enzymes, and AS CLEA, and their respective assignments are listed in Appendix A (Table A1). In general, the multiple peaks occurring between 1000 and 1700 cm^−1^ indicated different bending and stretching vibrations associated with C-O, C-C, C-N, C=O, and N-H linkages of enzyme molecules [50]. In particular, the peaks at 1628 cm^−1^ (which could be assigned to lysine NH^+^ and/or NH_3_^+^ asymmetric bending [51]) and 1505 cm^−1^ (which could be assigned to N-terminal, e.g., -NH_2_ and -NH_3_^+^ [52]) decreased in the CLEA when compared to the crude AS enzyme extract. By contrast, the peak at 1634 cm^−1^ (imine C=N stretching) increased in the CLEA, thus confirming the formation of the Schiff’s base linkages between the amino groups of the enzyme and the aldehyde groups of the cross-linker.

The appearance of a peak at 1520 cm^−1^ (Figure 3) in the cross-linked sample compared to the crude enzyme can be attributed to the formation of positively charged, quaternary-amino groups [53]. The peak representing amide/amine C-N and N-H shifted from 1245 cm^−1^ in the crude enzyme extract to 1235 cm^−1^ in the CLEA (Figure 3), which was probably due to the conformational sensitivity of these chemical bonds [54]. Chaudhari and Singhal [50] similarly observed shifts in the amide bands of the FTIR spectra of chitosan and cutinase cross-linked with glutaraldehyde.

### 3.4. Hydrolysis Kinetics of Wheat Flour

The hydrolysis of the wheat flour and the release of the sTOC, reducing sugar, and tyrosine by AS CLEA addition (prepared using optimum reaction conditions, see Section 3.2.2) was described by a first-order reaction model (Equation (3)) during the initial 6.5 h incubation period with a high level of statistical confidence (*p* < 0.05, R^2^ > 0.96; specific model coefficients are displayed in Table 1, and the relationships are shown in Figure 4). The enzymatic hydrolysis reaction velocity for the sTOC, reducing sugar, and tyrosine accumulation, indicated by *k*, was 2.3, 1.2, and 2.3 times higher, respectively, at the CLEA addition rate = 1:10 *v*/*v* compared to the CLEA addition rate = 1:100 *v*/*v*. The maximum potential concentration of products, *C_∞_* (Equation (3) and Table 1), provides a general indication of the total yield (due to the absence of experimental results in the asymptote region); nevertheless, the maximum yields (estimated by Equation (4)) of the sTOC, reducing sugar, and tyrosine from the hydrolysis of the wheat flour at CLEA rates of 1:100 and 1:10 *v*/*v* were 8.0 and 9.3%, 4.1 and 6.5%, and 75.9 and 108.7%, respectively. The results from the kinetic model (Figure 4) suggested that 90% of the equilibrium concentration (*C*_∞_) of the hydrolysis products may be potentially achieved after a reaction period of 26 h at both of the rates of CLEA addition. However, the experimental samples at 24 h and 26 h were less than the predicted values (and were, therefore, excluded from the model), which was possibly due to interference from the consumption of the products of enzymatic hydrolysis by other microbial reactions in the system [55]. While it is desirable to include the data in the asymptote range, the initial data (0–6.5 h) showed a consistent pattern of reaction kinetics and product accumulation (Table 1); therefore, the model provided indicative values for the rate of the enzymatic reactions and the total hydrolyzed fraction.

The accumulation of the reducing sugar and tyrosine demonstrated the effective hydrolysis of the starch and protein contained in the wheat flour by the AS CLEA. Butterworth et al. [40] reported a reducing sugar yield of 11.7% from the hydrolysis of wheat starch (5 g/L TS) using a commercial amylase enzyme (dose = 0.33 U/mL). In this study, the indicative reducing sugar yield obtained for the AS CLEA was moderately smaller, at 4.1 and 6.5% for CLEA dose rates of 1:100 and 1:10 *v*/*v*, respectively, which may be explained by the reduced amylase activity in the CLEA, equivalent to 0.030 and 0.27 U/mL, respectively, compared to the commercial enzyme formulation.

In contrast, the tyrosine showed a much greater yield compared to the reducing sugar, which was equivalent to 75.9% and 108.7% at CLEA rates of 1:100 and 1:10 *v*/*v* respectively, even though the protease activity was much smaller compared to the amylase, which was equivalent to 0.0010 U/mL and 0.0094 U/mL, respectively. The varying behaviors of hydrolytic enzymes against different substrates may be explained by the dynamic chemical interactions that occur between polymer molecules in complex matrices. For example, proteins can physically adsorb onto starch granules to form a cross-linked network in a flour suspension [56], thereby coating the starch granules and restricting the access of the amylase to the starch–glycosidic bonds. Protein molecules are, therefore, located on the outside of starch granules and, consequently, are more accessible and susceptible to protease attack. Glucose from the enzymatic hydrolysis of starch and non-hydrolyzed starch residues (such as linear-chain glucans) from wheat flour can also bind to the active sites on amylase [57] or become inter-tangled with adjacent glucans, protein, and fiber in the flour suspension [58], thus further hindering the hydrolysis of the starch in the flour by amylase. Overall, therefore, the patterns of starch and protein hydrolysis in the wheat flour by AS CLEA were generally comparable to commercial enzyme systems and were consistent with, and may be explained by, the dynamic interactions with substrate molecules present in complex organic matrices, such as wheat flour.

### 3.5. Hydrolysis Kinetics of γ-Irradiated WAS and Sonicated WAS

The enzymatic hydrolysis of both types of prepared WAS by CLEA addition was demonstrated through the release of tyrosine in the reaction mixture. However, the product concentration ultimately decreased towards the end of the incubation, and it was not possible to quantify the equilibrium value directly in the experiment, which was similar to the behavior observed for the wheat flour. This was presumably explained by microbiological activity and product consumption during the reaction period, which was anticipated, although the measures taken to limit the product degradation were only moderately successful. Nevertheless, the initial accumulation of tyrosine was effectively described by the first-order kinetic relationship (Equation (3)) and provided an indicative equilibrium concentration value: *C*_∞_; the model coefficients are displayed in Table 2, and the relationships are shown in Figure 5. The maximum potential yield of tyrosine from the hydrolysis of γ-irradiated and sonicated WAS by the CLEA was, therefore, calculated using Equation (4), and it was equivalent to 26.3% and 37.4%, respectively. By contrast, no accumulation of reducing sugar was observed for either type of WAS treatment (data not shown), thus suggesting that the CLEA protease was more effective in hydrolyzing the complex organic substrates, such as WAS, compared to the poly-saccharide-degrading enzymes in the CLEA (mainly amylase and cellulase). Indeed, these results are consistent with the greater effectiveness of protease at degrading the EPS fraction of bacteria biofilms reported by Molobela et al. [59] compared to amylases, which had corresponding hydrolysis rates at 26 °C in the range 36% to 75% and 9% to 28%, respectively.

The apparently limited effectiveness of poly-saccharide-degrading enzymes in the CLEA compared to protease at WAS hydrolysis may be explained by the following reasons: (1) poly-saccharide–protein interactions (similar to the starch–protein interaction discussed in Section 3.4) facilitated CLEA protease attack, but they limited the access and availability of the poly-saccharides to enzymatic hydrolysis, (2) the poly-saccharide structure of the EPS in WAS was resistant to poly-saccharide-degrading enzymes [59], and (3) protein was the dominant constituent fraction of the AS matrix compared to poly-saccharides [60].

Thus, the enzymatic hydrolysis of the whole WAS by the CLEA (mentioned in Section 2.7.3, data not shown) was probably limited by the protective effects of the EPS in WAS microbial cell floc structures and bacterial cell walls. Therefore, CLEAs could be an effective pre-treatment in combination with, and following, other chemical/physical methods to remove protective EPSs and disrupt the sludge prior to enzymatic hydrolysis.

## 4. Conclusions

The preparation of CLEAs from WAS produced by biological wastewater treatment has been reported for the first time. The optimum conditions for CLEA production were acetone precipitation (volume ratio = 4:1, 0.5 h at 0 °C), followed by dextran aldehyde cross-linking (0.04% *w*/*v*, 3.5 h at room temperature) with 5 mg/mL of BSA addition. The maximum recoveries of the amylase, protease, lipase and cellulase relative to the crude enzyme extract were 42.6%, 48.3%, 54.4% and 35.1%, respectively. The contrasting yields of reducing sugar and tyrosine from the hydrolysis of complex substrates, including wheat flour and AS, by the CLEA suggested that the performance of enzymatic hydrolysis also depended on the structural nature of the macro-molecules in the matrix (e.g., the poly-saccharide–protein interactions, and poly-saccharide structure). Nevertheless, the overall results demonstrated that AS CLEA has significant potential as an industrial enzyme product for the hydrolysis and bioconversion of complex organic substrates. Possible applications of the AS CLEA for the hydrolysis of organic waste residues to produce valuable materials could thus include food-processing waste, which is rich in carbohydrates (cellulose, starch, and mono-saccharides), proteins, and lipids, and has significant potential for enzymatic transformation into an array of high-value products and lignocellulosic materials, which are complex heterogeneous natural composites that comprise three main bio-polymers, including lignin, celluloses, and hemicelluloses, which can be converted into valuable bio-products such as rare sugars, surfactants, and bio-fuels by multi-enzymatic treatment. However, the effectiveness and behavior of hydrolytic enzymes also strongly depend on the substrate chemical and structural environment; therefore, further research is required to optimize the operational enzymatic process conditions and identify potential substrate factors that impact enzyme activity, such as shielding catalytic sites. Understanding and identifying the specific microbial species responsible for producing the hydrolytic enzymes in this study could also assist in optimizing the preparation of AS CLEAs and their potential applications. Enzyme recovery, therefore, has the significant potential to diversify sludge management methods, generate economic benefits, and enhance the overall sustainability of municipal wastewater treatment systems operated by the water industry.

## Figures and Tables

**Figure 1 microorganisms-11-01902-f001:**
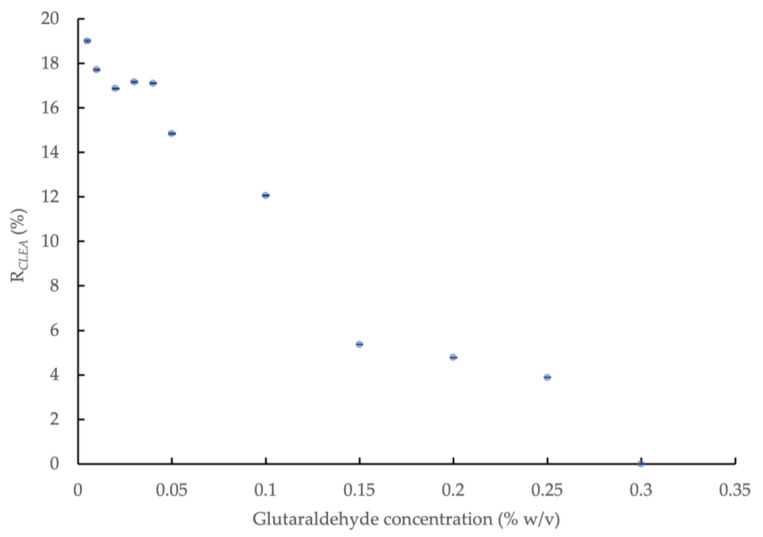
Effect of glutaraldehyde concentration on amylase activity of CLEAs prepared from AS extracts with 5 mg/mL BSA and precipitation with 80% saturated (NH_4_)_2_SO_4_ (*R_CLEA_*: volume activity in original enzyme solution was set to 100%).

**Figure 2 microorganisms-11-01902-f002:**
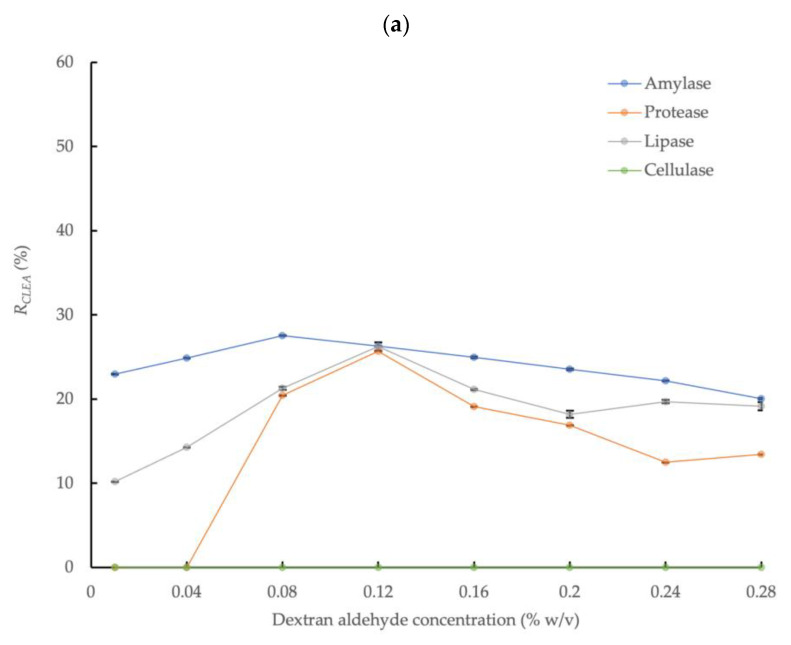
Effect of dextran aldehyde cross-linker concentration on activities of four different enzymes in CLEAs prepared from AS extracts with 5 mg/mL BSA and precipitation with (**a**) 80% saturated (NH_4_)_2_SO_4_ and (**b**) 4:1 acetone (*R_CLEA_*: volume activity in original enzyme solution was set to 100%; the cellulase sample at 0.16% *w*/*v* with acetone was compromised and the result is not reported; error bars represent the standard deviation).

**Figure 3 microorganisms-11-01902-f003:**
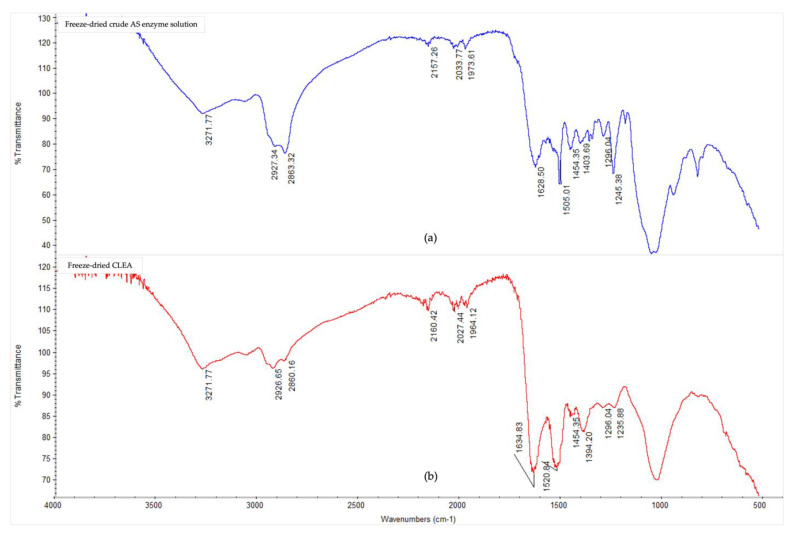
FTIR spectra in the 4000–500 cm^−1^ region of (**a**) freeze-dried crude AS enzyme solution, and (**b**) freeze-dried CLEA prepared under optimum conditions (acetone precipitation, BSA addition, and dextran aldehyde cross-linking (see Section 3.2.2)).

**Figure 4 microorganisms-11-01902-f004:**
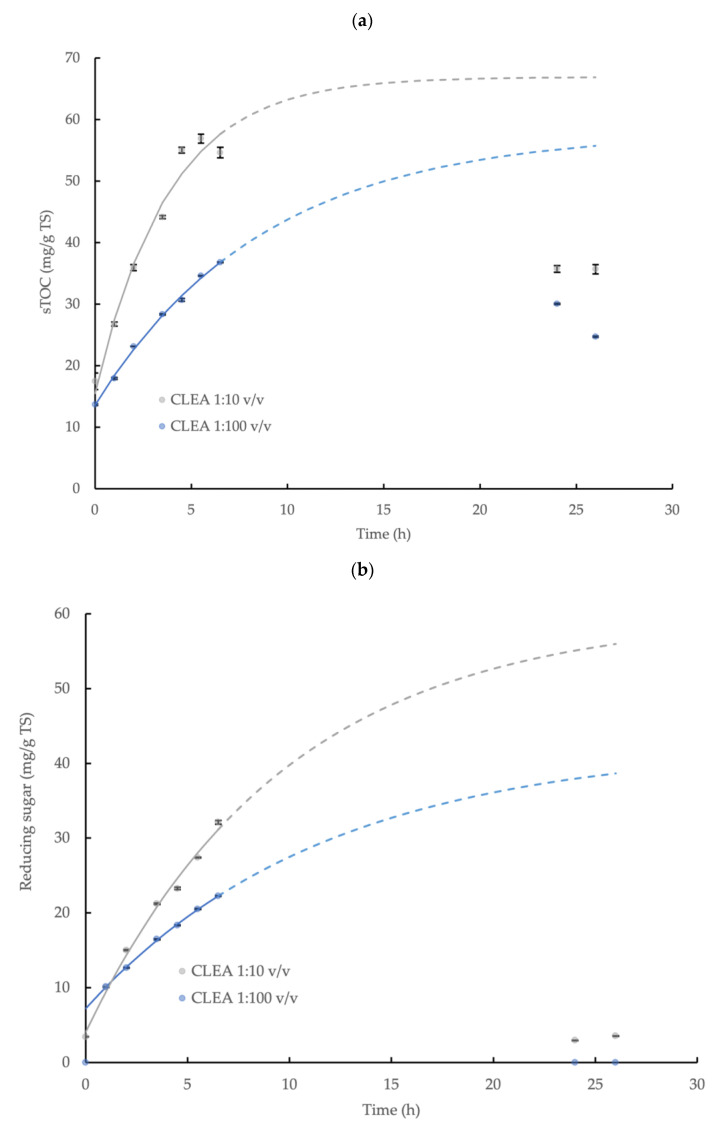
First-order kinetic model of (**a**) soluble total organic carbon (sTOC), (**b**) reducing sugar, and (**c**) tyrosine accumulation from the enzymatic hydrolysis of wheat flour by AS CLEA at two addition rates (1:100 and 1:10 *v*/*v*) and incubated at 40 °C, which show the maximum potential yield of each product (solid line represents model prediction based on experimental data within the 0–6.5 h incubation period (0–5.5 h for tyrosine at 1:10 ratio)); dashed line represents model prediction extrapolated to 26 h incubation period; error bars show the standard deviation).

**Figure 5 microorganisms-11-01902-f005:**
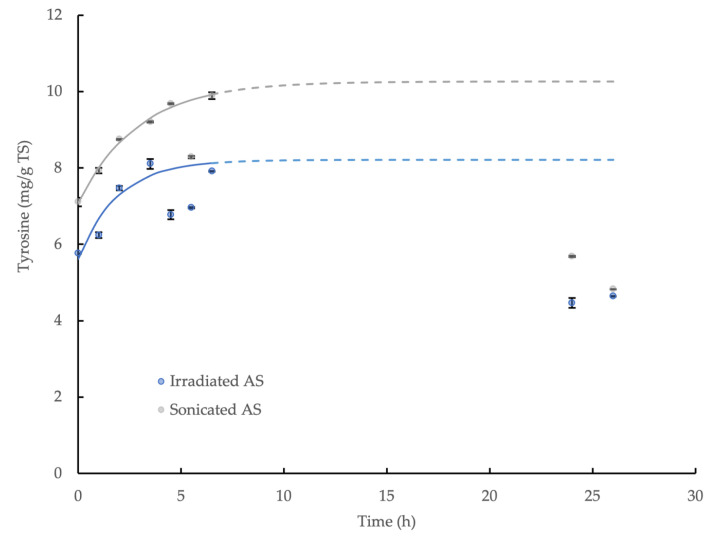
First-order kinetic model of tyrosine accumulation from the enzymatic hydrolysis of γ-irradiated WAS and sonicated WAS by AS CLEA at an addition rate of 1:10 *v*/*v* and incubated at 40 °C (solid line represents model prediction based on experimental data within the 0–6.5 h incubation period; dashed line represents model prediction extrapolated to 26 h incubation period; error bars show the standard deviation).

**Table 1 microorganisms-11-01902-t001:** First-order kinetic models of soluble total organic carbon (sTOC), reducing sugar, and tyrosine accumulation from the enzymatic hydrolysis of wheat flour by AS CLEA at two addition rates and incubated at 40 °C for 6.5 h (best-fit coefficients in 95% confidence interval ± standard deviation).

Chemical Parameter, Model Coefficient and Statistics	CLEA Dose Rate (CLEA Suspension: Substrate)	
	1:100 *v*/*v*	1:10 *v*/*v*
sTOC:		
*C*_0_ (mg/g TS)	13.6 ± 0.22	15.5 ±2.16
*C*_∞_ (mg/g TS)	58.0 ±3.78	66.9 ± 5.08
*k* (h^−1^)	0.114 ± 0.014	0.263 ± 0.059
Equation	*C* = 13.60 + 44.42 (1-e^−0.114t^)	*C* = 15.51 + 51.39 (1-e^−0.263t^)
R^2^	>0.99	0.96
*p*-value for normality test	>0.05	>0.05
Pass normality test (alpha = 0.05)	Yes	Yes
Reducing sugar:		
*C*_0_ (mg/g TS)	7.19 ± 0.12	4.05 ± 0.45
*C*_∞_ (mg/g TS)	42.4 ± 2.31	59.9 ± 9.39
*k* (h^−1^)	0.086 ± 0.008	0.102 ± 0.024
Equation	*C* = 7.19 + 35.24 (1-e^−0.086t^) *	*C* = 4.05 + 55.82 (1-e^−0.102t^)
R^2^	>0.99	0.99
*p*-value for normality test	>0.05	>0.05
Pass normality test (alpha = 0.05)	Yes	Yes
*Tyrosine*:		
*C*_0_ (mg/g TS)	0.92 ± 0.02	1.34 ± 0.02
*C*_∞_ (mg/g TS)	4.47 ± 0.34	5.97 ± 0.08
*k* (h^−1^)	0.125 ± 0.018	0.287 ± 0.010
Equation	*C* = 0.92 + 3.55 (1-e^−0.125t^)	*C* = 1.34 + 4.63 (1-e^−0.287t^) **
R^2^	>0.99	>0.99
*p*-value for normality test	<0.001	>0.05
Pass normality test (alpha = 0.05)	No ***	Yes

Note: *, fitting of the model was based on data within 1–6.5 h, since reducing sugar was not detected at t = 0; **, fitting of the model was based on data within 0–5.5 h, results measured at 6.5 h were excluded due to the inconsistent response compared with the previous pattern of kinetic results (formation of sediments was observed in the sample at 6.5 h (data not shown), probably due to self-aggregation of the proteins or protein aggregation with other complex polymers in the wheat flour; the apparent decline in tyrosine may therefore be explained by the Lowry method (specified in Section 2.4), which quantifies soluble products of protein hydrolysis); ***, the residual did not pass the normality test; therefore, a visual examination of the residual values (data not shown) was carried out (Section 2.7.4 and Appendix B), which showed a small systematic deviation from the first-order kinetic model; however, although statistically significant, this was relatively minor, as indicated in Figure 4c, and generally less than 4% of the predicted values for individual replicates.

**Table 2 microorganisms-11-01902-t002:** First-order kinetic model of tyrosine accumulation from the enzymatic hydrolysis of γ-irradiated WAS and sonicated WAS by AS CLEA at an addition rate of 1:10 *v*/*v* and incubated at 40 °C for 6.5 h (best-fit coefficients in 95% confidence interval ± standard deviation).

Sludge Type	γ-Irradiated WAS	Sonicated WAS
*C*_0_ (mg/g TS)	5.61 ± 0.18	7.08 ± 0.06
*C*_∞_ (mg/g TS)	8.21 ± 0.23	10.27 ± 0.12
*k* (h^−1^)	0.521 ± 0.131	0.342 ± 0.031
Equation	*C* = 5.61 + 2.60 (1-e^−0.521t^) *	*C* = 7.08 + 3.19 (1-e^−0.342t^) **
R^2^	0.91	>0.99
*p*-value for normality test	>0.05	>0.05
Pass normality test (alpha = 0.05)	Yes	Yes

Note: *, model fitting based on data obtained within 0 ≤ t ≤ 3.5 h and t = 6.5 h; **, model fitting based on data obtained within 0 ≤ t ≤ 6.5 h, excluding data at t = 5.5 h.

## Data Availability

Data is available upon request.

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
