# Peer review of "Cross-Linked Enzyme Aggregate (CLEA) Preparation from Waste Activated Sludge"

_microorganisms, 2023, doi:10.3390/microorganisms11081902_

Round 1

Reviewer 1 Report

The text that is the subject of this review is devoted to the study of obtaining and properties of enzymes from activated sludge. The topic is important from both a cognitive and practical point of view.

The text is of great cognitive value, especially regarding the kinetics of the enzymatic reaction of starch hydrolysis using a mixture of the abovementioned enzymes.

However, during a detailed reading, some ambiguities may appear in the reader's mind that require further clarification. Here are some of them:

-        It is a pity that the authors did not identify the microorganisms from the activated sludge used in the study. Such knowledge would give significant cognitive.

-        By precipitation of enzyme mix with acetone, a volume ratio of acetone: enzyme mix 4:1 was used - why such an arbitrary value?

-        In L508-509, it is stated that the optimal conditions are "volume ratio = 4:1, 0.5 h at 0 °C". Were studies conducted with different volume ratios of acetone to the enzyme mixture at different temperatures and times?

-        In Figure 2, the RCLA value for the cellulase course of the curve for concentrations of dextran aldehyde of 0.16 and 0.2 is incomprehensible - what causes such glaring instability? Is this a measurement error or some other methodological error?

-        Figure 4 shows the hydrolysis kinetics of wheat flour. In L426, it is written that the results for the time 24 and 26h have been omitted, and in L426 that "there is desirable to include data in the asymptote range". So were these values ​​finally omitted? And what about the results from Figure 4b for 6.5h (1:10 v/v) - why such a glaring deviation from the rest of the data?

-        In Figures 4 and 5, error bars, which should show standard deviation, are almost invisible – they are drawn behind the points and should be placed in front of them.

-        Description of Figures 2(a) is on the next page, as are Figures 4(a) and 4(b) - this makes it difficult to read the text.

-        The conclusions are too concise. They should be expanded, in particular indicating practical applications.

Author Response

We thank the reviewer for the valuable feedback. Please see the attachment for the response to your comments and suggestions. Please also see the revised manuscript, where all revisions to the manuscript were marked up using the “Track Changes” function as requested by the editor. Many thanks.

Reviewer 2 Report

This research is well-designed and add to the existing literature. Major comments:

1 - Highlight the study's novelty. The introduction should present the findings of similar works and show how the present research will add to the existing literature. 

2  - Main aim presenting in the introduction is clear and enough, I would not add specific objectives in the introduction sentence.

3  - The quality of fig. 3 and 1 (supporting content) must be enhanced. Please, check figs' titles.

4  -References must be updated and checked. Several of them are too old and could be replaced. Also, I am not sure if some information were correctly referenced. For example:

"2) apply the AS CLEA to the hydrolysis of complex organic  substrates, including wheat flour, as a representative complex organic substrate source, and waste activated sludge (WAS) from the biological wastewater treatment process, as a potential pretreatment technique, for example, to anaerobic digestion [15]."

15. Wingfield, P. Protein precipitation using ammonium sulfate. Current protocols in protein science 2001, Appendix 3, Appendix

The authors are advised to read recent studies, such as https://doi.org/10.1016/j.jwpe.2020.101857; https://doi.org/10.3390/app10217763; https://doi.org/10.1016/j.biortech.2018.11.092.

Author Response

(The authors gave the same response as above.)

Round 2

Reviewer 2 Report

The authors have addressed all previous comments. The manuscript can be accepted in its current state.